# The CRISPR/Cas9 Minipig—A Transgenic Minipig to Produce Specific Mutations in Designated Tissues

**DOI:** 10.3390/cancers13123024

**Published:** 2021-06-16

**Authors:** Martin Fogtmann Berthelsen, Maria Riedel, Huiqiang Cai, Søren H. Skaarup, Aage K. O. Alstrup, Frederik Dagnæs-Hansen, Yonglun Luo, Uffe B. Jensen, Henrik Hager, Ying Liu, Henrik Callesen, Mikkel H. Vendelbo, Jannik E. Jakobsen, Martin Kristian Thomsen

**Affiliations:** 1Department of Clinical Medicine, Aarhus University, 8200 Aarhus, Denmark; mfb@biomed.au.dk (M.F.B.); maria.riedel@biomed.au.dk (M.R.); huiqiang.cai@clin.au.dk (H.C.); soerskaa@rm.dk (S.H.S.); aagealst@rm.dk (A.K.O.A.); uffejens@rm.dk (U.B.J.); 2Department of Respiratory Diseases and Allergy, Aarhus University Hospital, 8200 Aarhus, Denmark; 3Department of Nuclear Medicine & PET, Aarhus University Hospital, 8200 Aarhus, Denmark; mhve@biomed.au.dk; 4Department of Biomedicine, Aarhus University, 8000 Aarhus, Denmark; fdh@au.dk (F.D.-H.); ALUN@biomed.au.dk (Y.L.); JEJ@biomed.au.dk (J.E.J.); 5Lars Bolund Institute of Regenerative Medicine, BGI-Qingdao, BGI-Shenzhen, Shenzhen 518083, China; 6South Denmark Hospital, 7100 Vejle, Denmark; henrik.hager@rsyd.dk; 7Department of Animal Science, Aarhus University, 8830 Tjele, Denmark; ying.liu@anis.au.dk (Y.L.); henrik.callesen@anis.au.dk (H.C.); 8Aarhus Institute of Advanced Studies (AIAS), Aarhus University, 8000 Aarhus, Denmark

**Keywords:** CRISPR, cancer, animal models, porcine model, lung cancer, handmade cloning, TP53, STK11, KRAS, Göttingen minipigs

## Abstract

**Simple Summary:**

Research in large animal models has been hampered by the complexity to introduce new gene alterations, but this has been simplified by the discovery of the CRISPR/Cas system. Here, we have cloned a Cas9 minipig to generate a porcine model for pre-clinical research. Six viable piglets were produced and backcrossed to Göttingen minipigs for two generations. Primary cells from different organs were isolated, and multiple gene alterations were performed by CRISPR in vitro. In vivo activation of the Cas9 expression was conducted by viral delivery of the FlpO expression to the skin. Overall, we successfully cloned a Cas9-expressing minipig and confirmed gene alterations introduced by the CRISPR/Cas system to porcine cells.

**Abstract:**

The generation of large transgenic animals is impeded by complex cloning, long maturation and gastrulation times. An introduction of multiple gene alterations increases the complexity. We have cloned a transgenic Cas9 minipig to introduce multiple mutations by CRISPR in somatic cells. Transgenic Cas9 pigs were generated by somatic cell nuclear transfer and were backcrossed to Göttingen Minipigs for two generations. Cas9 expression was controlled by FlpO-mediated recombination and was visualized by translation from red to yellow fluorescent protein. In vitro analyses in primary fibroblasts, keratinocytes and lung epithelial cells confirmed the genetic alterations executed by the viral delivery of single guide RNAs (sgRNA) to the target cells. Moreover, multiple gene alterations could be introduced simultaneously in a cell by viral delivery of sgRNAs. Cells with loss of TP53, PTEN and gain-of-function mutation in KRAS^G12D^ showed increased proliferation, confirming a transformation of the primary cells. An in vivo activation of Cas9 expression could be induced by viral delivery to the skin. Overall, we have generated a minipig with conditional expression of Cas9, where multiple gene alterations can be introduced to somatic cells by viral delivery of sgRNA. The development of a transgenic Cas9 minipig facilitates the creation of complex pre-clinical models for cancer research.

## 1. Introduction

Breakthroughs in regards to diagnosis and treatment of diseases, including cancer, have often been obtained from studies in small animal models. Here, the mouse is the preferred model, highlighted by the number of models generated [1]. However, discoveries determined in murine disease models are often untranslatable to the clinic [2]. It has been demonstrated that animals with larger body mass and a higher number of cells have developed distinct mechanisms to avoid cancer, which are not found in the mouse [3]. A large disease model, such as the pig, could, thus, help to increase the success rate for bridging basic biomedical studies to clinical trials. Compared to mice, pigs are more similar to humans in the majority of biological aspects [4,5,6]. Porcine cancer models have been generated with limited success, compared to the large number of mouse models [7,8,9,10,11,12,13,14]. Single gene alteration has often not resulted in cancer, and transgenic pigs with overexpression of oncogenes have resulted in other complications that have compromised the model [13,14]. The few successful cancer models in pigs have revealed new aspects of cancer biology, for example highlighted by the insight to TP53, which is not evolutionary conserved to the mouse [15]. Hence, large animals can contribute to understanding cancer biology, leading to better cancer treatments.

The CRISPR/Cas9 system consists of the Cas9 endonuclease and single guide RNA (sgRNA) that must be present simultaneously to induce genetic modifications [16]. The CRISPR/Cas9 system induces mutations at the sgRNA target site by deletions or insertions (Indel) of base pairs, which often disrupt the reading frame [16]. The CRISPR/Cas9 system has been applied to in vivo studies and is widely used in mouse studies [16,17,18]. The method has multiple advantages, especially by simultaneous multiplex gene alteration in somatic cells. For example, lung cancer has been induced in mice by targeting *Trp53*, *Stk11* and *Kras*, followed by rapid cancer development [19,20].

Transgenic porcine disease models are generated by expensive and time-consuming methods such as pronuclear injection or somatic cell nuclear transfer (SCNT). Notably, cloning is inefficient and only a few percent of the implanted blastocysts/morulae will result in viable piglets [21,22,23]. In addition, an intercrossing of different porcine models is laborious due to the long generation time compared to rodents. Applying the CRISPR/Cas9 system to large animal models, such as pigs, could overcome some of the obstacles with generating new lines and study multiple gene alterations in vivo and in vitro.

Here, we cloned a minipig with inducible Cas9 expression that can be activated upon delivery of FlpO recombinase. Transgene expression was visualized by red fluorescent protein (RFP) expression. Activated, Cas9-expressing cells were visualized by a shift to expression of yellow fluorescent protein (YFP). Cas9-expressing minipigs were cloned by SCNT and backcrossed to Göttingen Minipigs for two generations. Two different clones were obtained with strong transgene expression levels across different organs. The isolation of fibroblast, keratinocytes and lung epithelial cells showed strong transgene expression and activation in vitro. Furthermore, successful gene alterations were performed by the delivery of sgRNA’s. Multiple gene alterations transformed the porcine cells in vitro and increased cell proliferation. In vivo activation of Cas9 expression was performed in the skin, validating the design of the construct in vitro and in vivo.

## 2. Materials and Methods

### 2.1. Animals

All experiments were carried out at Aarhus University and Aarhus University Hospital. The cloned pigs were mixed-race minipigs. F1 and F2 offspring were obtained by backcrossing with Göttingen Minipigs from Ellegaard Göttingen Minipigs, Dalmose, Denmark [24]. WT control pigs were age-matched Danish Landrace–Yorkshire pigs. Deeply sedated pigs were euthanized by intra-cardiac injection of 30% pentobarbital (1 mL pr. 10 kg body mass).

### 2.2. Vector Construction

The CRISPR-Cas9 transposon vector (LIR_TRE_CAG_mKateII-Puro-STOP_mVenus-Cas9-PA-RIR, Addgene no. 68345) was constructed over multiple cloning steps. In brief, an A2-linked Venus gene was added to the pX330-U6-Chimeric_BB-CBh-hSpCas9 plasmid (Addgene no. 42230), using the AgeI and BglII restriction sites. Next, the Venus-2A-Cas9 construct was transferred to the TRE VKCS_KRAB with the stopper kat_Puro plus ROX SV40PA_GOLD Global plasmid (Addgene no. 67277), using the NotI and AgeI restriction sites. Finally, Katushka was replaced with synthesized mKate2, using the PacI and AvrII restriction sites.

The PKT plasmid was generated by inserting a synthesized pig KRASG12D repair template and 3′ AAV2 ITR into the AAV_3xgRNA;PTENA, p53B,SMAD4A_CAG _FlPO_synthPA plasmid (Addgene no. 68346), using the RsrII and KasI restriction sites. Next, the SMAD4 sgRNA was replaced by a KRAS sgRNA that was inserted into the pSpCas9(BB)-2A-GFP (PX458) plasmid (Addgene no. 48138), using the ClaI and KpnI restriction sites.

The PTKN plasmid was generated by inserting the NOTHC1 sgRNA, which was inserted into the pSpCas9(BB)-2A-GFP (PX458) plasmid that was cloned into the PKT plasmid using the KpnI site.

The SKT plasmid was generated by replacing the PTEN and TP53 sgRNAs with a STK11 sgRNA that was inserted into the pSpCas9(BB)-2A-GFP (PX458) plasmid (Addgene no. 48138) using the NotI and ClaI restriction sites. Subsequently, the TP53 sgRNA was re-inserted using the KpnI restriction site.

### 2.3. Single Guide RNA (sgRNA) Design and Validation

The sgRNAs were designed using a CRISPR design tool. See Appendix A for sgRNA sequences and genomic primers. The guide efficacy was determined with the TIDE (Tracking of Indels by Decomposition) software through transfection of porcine fibroblasts with the pSpCas9(BB)-2A-GFP plasmid (Addgene ID: 48138), harbouring the designated sgRNA followed by FACS of GFP positive cells.

### 2.4. ICE Analysis

Sanger sequencing of PCR products from the target site of the guides was analysed by the ICE v2 CRISPR Analysis Tool to identify mutations.

### 2.5. Handmade Cloning (HMC), Culture and Transfer of Embryos

Porcine fibroblasts were cultured from ear biopsies of juvenile mixed-race minipigs as previously described [25]. Transfection with 400 ng LIR_TRE_CAG_mKateII-Puro-STOP_mVenus-Cas9-PA-RIR transposon plasmid and 50 ng hyperactive Sleeping Beauty transposase plasmid using the Turbofect Transfection Reagent according to the manufacture’s recommendations (Thermo Scientific, Roskilde, Denmark) generated the transgenic Cas9 fibroblasts. Positive cells were selected with puromycin (Sigma-Aldrich, Søborg, Denmark) for 8 days. The resistant colonies were analysed by assessment of the RFP fluorescent signal using a fluorescence microscope to determine their fluorescent profile. A pool of cells with different fluorescent intensity profiles were selected and grown for 9 days prior to SCNT by HMC, as described [26]. The reconstructed, transgenic embryos were cultured in vitro for 5 to 6 days, and selected blastocysts/morulae were surgically transferred to a Danish Landrace recipient sow. Cloned, transgenic piglets were delivered by natural farrowing on day 115 and raised by their surrogate mother. From the 85 blastulas/morulas transferred to the surrogate mother, 10 male minipigs were produced, of which 3 were stillborn.

### 2.6. Cell Work

Primary porcine fibroblasts were isolated from ear biopsies of neonatal transgenic piglets. Single cells were obtained by trypsin treatment and then expanded in DMEM containing 100 U/mL penicillin, 0.1 mg/mL streptomycin (Sigma-Aldrich, Søborg, Denmark), and 15% foetal bovine serum (Gibco, Roskilde, Denmark) at 37.5 °C.

Primary Cas9 keratinocytes were isolated from ear biopsies from neonatal Cas9 minipigs by bisecting the biopsies, whereby the epidermis was exposed. The explants were placed in culture flasks and incubated bottom-up overnight at 37.5 °C. Outgrowth and expansion of keratinocytes was achieved in DMEM supplemented with 15% foetal bovine serum (Gibco, Roskilde, Denmark), 100 U/mL penicillin, 0.1 mg/mL streptomycin, 265 mg/L L-glutamine, 10 ng/mL EGF (Invitrogen, Roskilde, Denmark), 50 µM gentamycin and 0.4 µg/mL hydrocortisone (Sigma-Aldrich, Søborg, Denmark) at 37.5 °C for 7 to 10 days. Subsequently, the growth medium was changed to a serum-free keratinocyte medium, K-SFM, with an epidermal growth factor and bovine pituitary extract (Gibco, Roskilde, Denmark).

Fibroblasts and keratinocytes isolated from transgenic Cas9 piglets were transfected with a plasmid expressing Flp Optimized (FlpO) [27] under a ubiquitous CAG promoter to activate Cas9 and Venus expression or AAV particles carrying designated sgRNAs and FlpO. For the cell proliferation assay, 25,000 cells were seeded and counted after 3 days. Three different clones were used.

### 2.7. Air–Liquid Interface (ALI) Culture of Porcine Tracheobronchial Epithelial Cells

Porcine Cas9 lung tissue resections were transported and incubated for 72 h in DMEM (Sigma-Aldrich, Søborg, Denmark) with 1× Antibiotic–Antimycotic (Gibco, Roskilde, Denmark) and 1× Gentamicin solution, 0.1 mg/mL protease and 0.01 mg DNase (Sigma-Aldrich, Søborg, Denmark). Porcine tracheobronchial epithelial cells were harvested by scraping off the cells with a scalpel, FCS was added to inactivate the protease, and the media containing the brownish film with basal epithelial cells were centrifuged for 5 min at 500× *g*. The pellets were washed in Hanks’ Balanced Salt solution, BBS (Sigma-Aldrich, Søborg, Denmark), centrifuged for 5 min at 500× *g* and resuspended in BEGM (Lonza, Copenhagen, Denmark) with 1× Antibiotic–Antimycotic. The resuspended cells were plated in petri dishes for 2 h, where the remaining fibroblasts bound to the surface and the epithelial cells stayed in suspension. Subsequently, the suspended cells were transferred to type I and III collagen-coated dishes for expansion. Expanded basal cells were detached by trypsin treatment, centrifuged for 5 min at 500× *g* in PneumaCult™-ALI Medium (STEMCELL Technologies, Grenoble, France) at a density of 105 cells/mL, followed by seeding the cells onto type IV collagen (Sigma-Aldrich, Søborg, Denmark)-coated Transwell permeable supports (24 mm, 0.4 μm polyester membrane, Corning, Wiesbaden, Germany). 1 mL of ALI medium was added to the lower reservoir, and 0.5 mL of cell suspension was added to the upper reservoir. In less than 7 days, at 37 °C in 5% CO_2_, when cells were confluent, an ALI was formed by removing the apical medium, and the cells were fed with medium only from the basal compartment. Cultures were maintained under ALI conditions for at least 21 days prior to treatment to ensure formation of a pseudo lung epithelium.

### 2.8. In Vivo Imaging Systems Analysis

Organ biopsies from Cas9 minipigs, transported in ice-cold PBS, were used for ex vivo IVIS analysis to detect a fluorescent protein signal, using an IVIS Spectrum Preclinical In Vivo Imaging System (Perkin Elmer, Copenhagen, Denmark). The RFP signal was detected by excitation at 570 nm and emission at 640 nm. Data were collected in the form of average fluorescence radiant efficiency ((photons/s/cm^2^/steradian)/(µW/cm^2^)) and analysed with the Living Image software (Caliper Life Science, Copenhagen, Denmark). Organ biopsies from an age-matched Danish landrace pig were used as negative controls.

### 2.9. Fluorescence-Assisted Cell Sorting

Cell sorting was carried out using a BD FACSAria III high-speed cell sorter. GFP or YFP positive cells were analysed at a velocity of 2000 cells/s and sorted into a culture plate with DMEM ranging from 1 to 2000 cells per well at 4 °C.

### 2.10. Southern Blot Analysis

Genomic DNA was isolated from the chorion villi lysis buffer treated Cas9 pig fibroblasts. Southern blotting was performed on 15 μg DNA from each piglet, digested with 5 µL XmaJI for 12 h (Thermo Scientific, Roskilde, Denmark) and spiked with 3 µL XmaJI for another 4 h of incubation, as described [28]. The RFP DNA probe (656 bp) was purified from restriction-cleaved plasmid DNA and labelled with 32P-dCTP EasyTide (Perkin Elmer, Copenhagen, Denmark) using the Prime-It Random Labeling Kit (Agilent Technologies, Glostrup, Denmark). The X-ray film was developed at −70 °C for 48 h.

### 2.11. AAV Production

An amount of 90% confluent HEK293T cells, seeded on 15 cm dishes in 22 mL DMEN with 10% FCS and 1× pen-strep, were transfected with 14 µg Adeno-Helper plasmid; 14 µg serotype 6, 9, or 2H22 capsid plasmid; 14 µg viral construct plasmid, and 2.2 µg GFP plasmid (serving as transfection control), using the branched MW 25,000 polyethylenimine (PEI) (Sigma-Aldrich, Søborg, Denmark, 408727) transfection reagent. The PEI mixture consisted of 350 µL PEI and 440 µL H_2_O. The DNA mixture consisted of 44 µg plasmid and 760 µL H_2_O. A total of 790 µL 0.3 M NaCl was added to both mixtures just before they were mixed. The transfection mixture was incubated for 10 min, followed by an addition of 3.1 mL transfection mixture to each dish. Cells were harvested 48–72 post transfection using a cell scraper, collected in 50 mL tubes, and pelleted by centrifugation for 10 min at 400× *g*. The pellets were resuspended in a total volume of 15 mL in PBS and lysed by 5 freeze–thaw cycles (−196 °C liquid nitrogen to 37 °C water bath). Lysed cells were centrifuged at 4000× *g* for 5 min, and the supernatant was transferred to a new 50 mL tube. Excess RNA and DNA was removed by benzonase (Sigma-Aldrich, Søborg, Denmark) treatment for 1 h at 37 °C to a final concentration of 50 u/mL. Next, the crude AAV prep was filtered through a 0.45 µm filter using a syringe and then a 0.20 µm filter. Finally, the AAV sample was concentrated by loading the sample on a 100 K protein concentrator (Thermo Scientific, Roskilde, Denmark), followed by centrifugation at 4000× *g* until the desired volume was reached. The AAV samples were titrated by qPCR analysis using a plasmid standard curve and 1000 to 100,000 fold dilutions of the AAV samples. The Brilliant III Ultra-Fast SYBR^®^ Green QPCR Master Mix reagent (Aqilent Techonologies, Santa Clara, CA, USA) was used according to the manufacture’s recommendations. See Appendix A for primers.

### 2.12. AAV Delivery to Pig Lungs

Delivery of AAV particles to the lungs of sedated pigs was facilitated by endobronchial spraying in a specified lung segment during a bronchoscopy or aerosolization of AAV particles with an Aerogen Pro nebulizer that was connected to a Dräger Primus anaesthesia machine. Hereby, 10^12^ vg/pig SKT_AAV2H22 particles were administered to 3 pigs, 10^12^ vg/pig of SKT_AAV2H22 and 5 × 10^11^ vg/pig of PKT_AAV9 to 3 pigs, and 10^12^ vg/pig of SKT_AAV2H22 and 10^12^ vg/pig of PKT_AAV9 to 7 pigs.

### 2.13. Construct Delivery to the Skin of the Pigs

The PTKN construct was delivered to the skin of the Cas9 minipigs by either injecting 10^12^ PTKN_AAV6 particles, 10^12^ PTKN_AAV6 particles followed by electroporation, or electroporation of 100 µL (1 µg/µL) PTKN plasmid using a Cliniporator (IGEA, Wetherby, UK). The electric pulses consisted of 1 high-voltage pulse at 400 V/cm, duration 100 μs, and 1 low-voltage pulse at 56 V/cm, duration 400 ms. There was a lack of 1000 ms between the pulses. PBS injections with subsequent electroporation (*n* = 3) served as negative controls. Each treated area was marked with 4 dots by tattoo ink stains to locate the treated areas at the end of the study.

### 2.14. Β-galactosidase Staining

Lung biopsies were fixed on ice for 30 to 120 min in 4% paraformaldehyde and 0.2% glutaraldehyde (Sigma-Aldrich, Søborg, Denmark). Next, the biopsies were washed three times in PBS and stained in 1 mg/mL X-gal, 2 mM MgCl_2_, 5 mM K_3_Fe(CN)_6_, 5 mM K_4_Fe(CN)_6_, and 0.02% NP40 dissolved in PBS (Sigma-Aldrich, Søborg, Denmark). This was followed by an in-dark incubation at 37 °C until a stain appeared. Finally, the biopsies were fixed in 4% paraformaldehyde, embedded in paraffin, sliced in 4 µ sections, and counterstained with nuclear fast red (Sigma-Aldrich, Søborg, Denmark).

### 2.15. Western Blotting

Cells were lysed in RIPA buffer with protease and phosphatase inhibitors (Sigma-Aldrich, Søborg, Denmark). The samples were sonicated, subjected to SDS-PAGE, and immunoblotted. The blots were blocked with 2.5% BSA (Sigma-Aldrich, Søborg, Denmark) or 5% dry milk (BD) in PBS (Sigma-Aldrich, Søborg, Denmark), containing 0.1% Tween20 (Sigma-Aldrich, Søborg, Denmark) prior to probing with the following primary antibodies: RFP (AB223), GFP (D5.1), Flag (M2), phosphor-p44/42 MAPK (SC-9101), P-Akt (S473), beta-actin (AC-15), and Vinculin (V9131). Appropriate horseradish peroxidase-conjugated secondary antibodies were used for development (Jackson ImmunoResearch, Ely, UK).

### 2.16. Histochemical Analysis

Tissues were fixed in 4% formaldehyde or paraformaldehyde overnight, embedded in paraffin and sliced in 4 µm sections. Antigen retrieval was performed at 100 °C in a citrate buffer, pH 6 for 20 min. Sections were blocked in 2.5% BSA in PBS with 0.1% Tween20 and RFP (AB223), or Ki67 (SP6) was used as the primary antibody. Anti-rabbit horseradish peroxidase-conjugated secondary antibody was used for development (Jackson ImmunoResearch). Counterstaining was performed with DAPI (Vector Laboratories, Burlingame, CA, USA).

### 2.17. Real-Time Quantitative PCR

The total RNA from the porcine cells was isolated using the High Pure RNA Isolation Kit (Roche). An amount of 50 ng total RNA was used per sample in combination with the Brilliant III Ultra-Fast SYBR^®^ Green QPCR Master Mix (Agilent Technologies, Glostrup, Denmark) (see Appendix A for primer sequences). Data were analysed with the ΔΔCT method and normalized to GAPDH.

### 2.18. PCR

PCR was performed on genomic DNA, isolated from chorion villi lysis buffer treated cells or ear biopsies. The Q5 High-Fidelity DNA Polymerase (NEB) was used according to the manufacture’s recommendations. See Appendix A for primers.

### 2.19. 18F-Fluorodeoxyglucose Positron Emission Tomography–Computed Tomograph (PET/CT) Scanning

Whole body 18F-FDG-PET/CT scans were carried out after a 12 h fast and under isoflurane. 1 h after IV injection of approximately 300 Mbq 18F-FDG, pigs were scanned on a Siemens Biograph Vision 600 integrated PET/CT system. Data were reconstructed using TrueX + TOF with a 440 × 440 matrix, 4 iterations, 5 subsets, a Gaussian filter, and 2 mm full with half maximum. An experienced nuclear medicine physician, using the Hybrid Viewer software (Hermes Medical Solutions, Stockholm, Sweden), analysed all images.

### 2.20. Statistics

An unpaired 2-tailed Student’s *t*-test was used for statistical analysis of the data. *p* values ≤ 0.05 were considered a statistically significant difference between compared groups. The data shown are representative from one out of three or more experiments.

## 3. Results

### 3.1. Design of a Conditional Cas9 Expression Construct

To generate a transgene minipig with global conditional Cas9 expression, we designed a Sleeping Beauty transposon to deliver the Cas9 gene into the genome of porcine fibroblasts. The conditional Cas9 construct contained a red fluorescent protein (RFP) [29] and 4 polyadenylation signals to terminate transcription flanked by two FRT sites. This design permitted FlpO inducible Cas9 expression. Adding the RFP gene to the construct made it possible to identify cells expressing the transgene. When activated, i.e., the Cas9 expressing cells, they could be visualized by the expression of yellow fluorescent protein (YFP) [30] as the genes were linked by a 2A linker peptide [31]. Strong expression of the transgenes was mediated by the CAG promoter [32] (Figure 1A).

### 3.2. Generation of a Transgenic Cas9 Minipig by Cloning

For cloning of the pigs, low passage Cas9 fibroblasts were generated. These fibroblasts were generated by being co-transfected with the aforementioned Cas9 transposon and the Sleeping Beauty transposase, followed by puromycin selection [28]. Single clones were obtained and screened based on their RFP expression and morphology. The selected clones were pooled and used to generate reconstructed embryos by HMC. In total, 85 blastulae/morulae were obtained. These were transferred to a recipient sow, which farrowed ten piglets, three of which were stillborn, and one died at the age of 5 weeks. The remaining six pigs were healthy and developed as expected.

To arbitrate the copy numbers of the transposon in the cloned pigs, ear biopsies were obtained to isolate primary fibroblasts. DNA from these fibroblasts was used for Southern blotting to determine the transposon copy number. Three different clones had been generated, resulting in three piglets with one copy, two piglets with two copies, and one piglet with six copies of the transposon (Figure 1B). The six-copy piglet showed weakness and had to be euthanized 5 weeks after delivery, due to weight loss. The pigs that contained one or two copies of the transposon were successfully used for the production of F1 and F2 generations by crossing with Göttingen minipigs.

The genetic design was validated by transfecting transgenic Cas9 fibroblasts with a FlpO and sgRNA-expressing plasmid or an empty plasmid. Cells transfected with an empty plasmid appeared red, and FlpO transfected cells showed the expected transition from red to yellow cells (appearing green in the applied filter) (Figure 1C). Moreover, a Western blot validated the termination of the RFP expression and the onset of Cas9 expression post transfection (Figure 1D). Overall, minipigs with conditional expression of Cas9 were cloned and viable. Moreover, the introduction of FlpO expression mediated the recombination and expression of Cas9 and YFP protein.

### 3.3. Characterization of the Transgenic Expression

Transgenic expression from a transposon can show varying expression profiles in different tissues and cell types due to the random integration. To analyse the expression from the transposon in vivo, tissue biopsies were collected from major organs from pigs containing one copy of the transposon. The expression for RFP was analysed by qPCR, Western blotting and IVIS scanning. The majority of organs had high levels of expression except from the brain, lymph nodes and spleen (Figure 2A,B and Appendix A). In-depth analysis by immunohistochemistry validated RFP expression, especially in epithelial cells, but also in muscles and hepatocytes (Figure 2C). Collectively, these analyses demonstrated that the transposon is expressed in the majority of organs with a strong expression in epithelial and muscles cells.

### 3.4. In Vitro Validation of CRISPR/Cas9-Induced Gene Alterations

To induce specific mutations, single guide RNAs were designed for the target genes and validated in porcine fibroblasts (Appendix A and Appendix A). A repair template was designed to introduce a KRAS^G12D^ gain-of-function mutation. The template was validated in fibroblasts by co-transfection with a sgRNA against KRAS and the KRAS^G12D^ repair template. Single clones were analysed for the introduction of the specific mutation, and clones with the desired KRAS^G12D^ point mutation were identified (Appendix A). These analyses confirmed the CRISPR/Cas9 induced gene alteration in porcine fibroblasts.

To further validate the genetic design, primary cells isolated from the transgenic Cas9 piglets were transduced with different viral constructs and appropriate AAV serotypes. All viral constructs contained the FlpO gene and the KRAS^G12D^ repair template combined with different combinations of sgRNAs (Figure 3A). Three viral constructs were produced: the first containing sgRNAs against STK11, KRAS, and TP53 (SKT), the second containing sgRNAs against PTEN, KRAS, and TP53 (PKT), and the third containing sgRNAs against PTEN, TP53, KRAS, and NOTCH1 (PTKN). Primary fibroblast or keratinocytes were transduced with the three different AAV particles, and the transition from red to yellow fluorescent was detected, indicating the expression of FlpO from the viral constructs (Figure 3B). Furthermore, fibroblasts transduced with the viral constructs demonstrated a significantly increased proliferation rate compared to non-activated fibroblasts (Figure 3C). Moreover, PKT treated fibroblasts revealed a significant higher proliferation rate than SKT treated fibroblasts. An insertion and deletion (Indel) analysis validated the mutation of the targeted genes and the introduction of the KRAS^G12D^ point mutation (Appendix A). Western blotting confirmed the transition from RFP to YFP expression, the mutation of the targeted genes by increased phosphorylation of ERK due to the point mutated KRAS^G12D^, and the phosphorylation of AKT in PKT treated cells as a consequence of the PTEN mutation (Figure 3D).

Two of the viral constructs, STK and PKT, were generated to induce lung cancer in the Cas9 transgene pigs. To validate the viral construct in lung epithelial cells, an ex vivo lung culture assay was established of basal tracheobronchial epithelial cells. Transduction of the lung epithelial cells revealed an expansion of transduced cells surrounded by non-treated cells. This shows that the transformed cells could overcome contact inhibition from their surroundings (Figure 3E). In summary, transduced cells demonstrated a proliferation gain and that they overcame contact inhibition. These data reveal the validity of the genetic design in vitro and its potential for the induction of cancer by an increased proliferation and an overcoming of contact inhibition.

### 3.5. In Vivo Activation of Cas9 Expression by FlpO-Mediated Recombination

The skin of the pig shows high similarity to humans and can, thus, serve as a comparable model. To demonstrate in vivo recombination and activation of Cas9 expression, the skin of Cas9 minipigs was investigated. The delivery of sgRNAs and FlpO was achieved by intradermal injections of PTKN_AAV6 particles or PTKN plasmid, followed by electroporation or only PTKN_AAV6 transduction (Figure 4A). Skin biopsies were taken 1 month post treatment and a histology analysis was performed. The histology and proliferation were similar between normal skin and samples that received sgRNAs a and FlpO-expressing construct (Figure 4A). To assess the samples for the presence of the viral construct, a PCR analysis was performed. All biopsies from the AAV-transduced skin were positive for the presence of the viral vector. Two biopsies from the electroporated areas were positive, and the combined delivery of AAV and electroporation had one positive sample (Figure 4B). CRISPR/Cas9-induced mutations were analysed by Sanger sequencing, but no mutations were detected by this method. To confirm the recombination at the FlpO site and hereby the activation of Cas9 expression, a transition PCR analysis was performed (Figure 4B). An FlpO mediated excision of the stop cassette was confirmed in two biopsies, and Sanger sequencing validated the recombination (Figure 4C). To ensure that Cas9 protein was expressed, the biopsies were analysed by Western blotting which revealed an expression of the Cas9 protein (Figure 4D). Overall, these analyses confirm that Cas9 protein can be expressed in vivo by the delivery of FlpO.

### 3.6. Induction of Lung Cancer

To assess an in vivo gene alteration by CRISPR/Cas9 and ultimo the induction of cancer, lung tissue was transduced by AAV particles. Prior to the induction of lung cancer, the delivery of viral particles to the lung epithelium had to be addressed. Pilot studies demonstrated that reporter viruses with serotype AAV2H22 or AAV9 transduced pig lung epithelium by assessing β-galactosidase staining and immunohistochemical staining for GFP, respectively (Appendix A). Hereafter, 13 Cas9 minipigs were infected with 10^12^ vg of SKT_AAV2H22 alone or SKT_AAV2H22 [33,34] and PKT_AAV9 in combination. The viral particles were delivered as aerosols or with a bronchoscope to the lungs. The pigs were monitored for cancer development by FDG-PET/CT-scans up to three times over a period of 18 months (Figure 5A). Three of the pigs showed minor abnormalities in the lungs by FDG-PET/CT scanning. One had an opacity in a part of a lobe. At termination, biopsies were analysed by histology and revealed features of inflammation (Figure 5B), which is fully compatible with the scans. One pig had a dense spot with a diameter of 5 mm at 8 months post transduction. After 13 months, an increased FDG-uptake was observed in this spot compared to the background—an indication of increased metabolism (Figure 5C). At termination, the foci were identified and revealed a calcification deposit. Multiple samples from different pigs were analysed for CRISPR-induced mutations by Sanger sequencing, but this method could not detect any alterations induced by the sgRNAs. In summary, these analyses revealed that targeting up to three genes simultaneously by AAV delivery of sgRNA to the lung tissue was not sufficient to induce cancer over a period of 18 months.

## 4. Discussion

Only a few models of cancer have been generated in large animals, as most models are based on rodents. The establishment of large animal models is an unmet need, as they have, in many aspects, a much higher similarity to humans [4,5,6]. Though, humans, as well as pigs, are less predisposed to develop cancer than mice [35]. In this study, we cloned a minipig with a Cas9-expressing construct controlled by FlpO recombinase. By an introduction of Cas9 to the porcine genome, it is possible to genetically alter genes by delivering sgRNAs. We have shown that different primary cells derived from the Cas9 minipig can be modified by CRISPR in vitro. Furthermore, the in vivo activation of Cas9 by a recombination through the expression of FlpO was performed in the skin. Hereby, we have applied the CRISPR/Cas system to the porcine cells, allowing for genetic alteration by specific sgRNAs.

In line with our Cas9 minipig, Wang et al. generated a CRISPR/Cas9 pig. Here, Wang et al. showed both, in vitro and in vivo gene alteration introduced by sgRNAs, which resulted in lung cancer formation [36]. They infected two 1 week old transgenic pigs with 5 × 10^6^ lentiviral particles and observed lung cancer 3 months post treatment. This shows that it is possible to generate lung cancer in pigs by the CRISPR/Cas9 technology. In contrast to our study, they relied on the Cre-LoxP system to induce the Cas9 expression [37,38]. They used a lentivirus as a vector, and they introduced mutations to BRCA1 BRCA2, APC, PTEN, TP53, and KRAS. We delivered 10^12^ AAV particles but did not observe lung cancer after 18 months. The lentivirus is integrated into the genome, unlike the AAV, which could accelerate cancer formation. Cancer induction in their pig model resembles tumour formation in a similar Cas9 mouse model, where we and others have shown cancer formation 3 months post transduction with AVV particles containing sgRNAs [19,20]. Multiple discrepancies between their and our model could explain the different outcomes. One may be related to the strain background of the used pigs or the time of delivery of sgRNAs. This could explain a later unset of lung cancer but is less likely to explain the lack of cancer formation, as the pigs were followed for 18 months. An analysis of the transgene expression from our transgenic pig varied among organs and even within tissues. A comparison of the tissues was compromised by a lack of uniform house-keeping genes. Immunohistochemistry for transgene expression revealed mosaic expression. In the lung, we estimated the expression of the transgene to be present in 25% of the epithelial cells. Therefore, we used additional amounts of AAV particles, to ensure the transduction of the Cas9-expressing cells. Future work will share light on this discrepancy and improve the robustness of cancer models in pigs.

The frequency of cancer in larger animals is similar or lower when compared to small animals, suggesting that large animals have evolved additional tumour suppressor functions [3,35]. We have compared mouse with porcine models by targeting the same genes in the pig that introduced lung cancer in mice by the use of the CRISPR/Cas9 technology [19,20]. The lack of cancer formation in the pig suggests a significant discrepancy between both models. At this stage, it remains unclear if this is related to biological differences or to external factors. One potential difference is the infection status in housing facilities for mice and pigs, which may affect the immune response towards the AAV, Cas9, or tumour surveillance. FDG-PET/CT-imaging revealed minor abnormalities in the lungs of pigs treated with the viral constructs by AAV transduction. A post-mortem examination of the pigs 18 months after the treatment identified these abnormalities as calcified deposits with no indication of mutations in target genes. In both, humans and pigs, foci with calcified deposits are observed as remains of immune responses against pathogens or sarcoidosis. These deposits originate from granulomas that harden over time [39,40]. The calcified foci we observed in our pigs could originate from our AAV treatment, since we also observed chronic inflammation in some pigs. This would explain why we did not observe the expected cancer formation, as the increased immune surveillance could have cleared transformed cells. FDG-PET/CT scanning of the pigs revealed activated lymph nodes, especially in the lungs and the colon (Appendix A), indicating an activated immune system, as it has been observed in other porcine models of cancer [41]. Together, these data indicate that a strong immune response can be induced in pigs, which could lead to cancer immune surveillance.

## 5. Conclusions

In conclusion, we developed a porcine model with inducible Cas9 expression in the majority of organs. We validated the model in vitro in fibroblasts, keratinocytes, and pseudo-stratified lung epithelial cultures and in vivo in the skin of the minipigs by Cas9 construct activation through AAV transduction. Thus, an in vitro and, potentially, an in vivo model of human cancer has been generated in a minipig for pre-clinical studies.

## Figures and Tables

**Figure 1 cancers-13-03024-f001:**
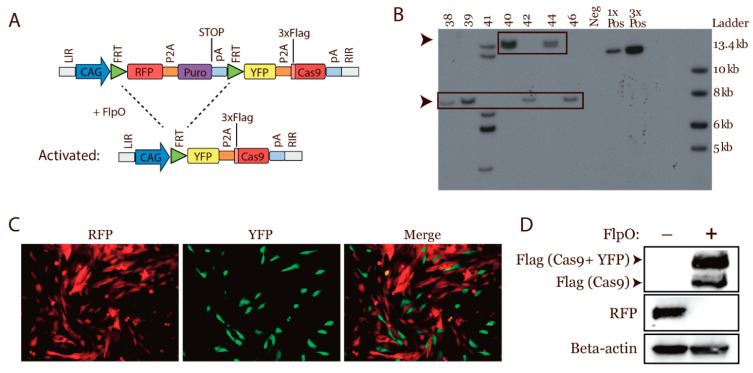
Generation of a transgenic Cas9 minipig. (**A**) Design of the transgenic FlpO-dependent CRISPR/Cas9 transposon: The CAG promoter drives the expression of the transgene, which initially mediates the expression of the RFP gene and the puromycin selection marker (Puro). The polyadenylation signal (pA) prevents transcription of the downstream YFP and Cas9 genes prior to FlpO-mediated activation. Both gene pairs are separated by 2A linker peptides (P2A). FlpO recombinase mediates construct activation by recombination at the FRT sites. This yields YFP and Cas9 expression. Cas9 protein contains a Flag tag. (**B**) Southern blot for the transgene copy number in seven cloned piglets. The arrowheads mark the piglets with one or two copies of the transposon. Pig 41 had six copies of the transposon. (**C**) Transition from RFP to YFP expressing cells by transfection with the FlpO recombinase. Images were taken after two passages. (**D**) Western blot on cell extracts seven passages after transfection with the FlpO or empty plasmid. Beta-actin was used as loading control. Western Blot and Southern Blot Images can be found in Appendix A.

**Figure 2 cancers-13-03024-f002:**
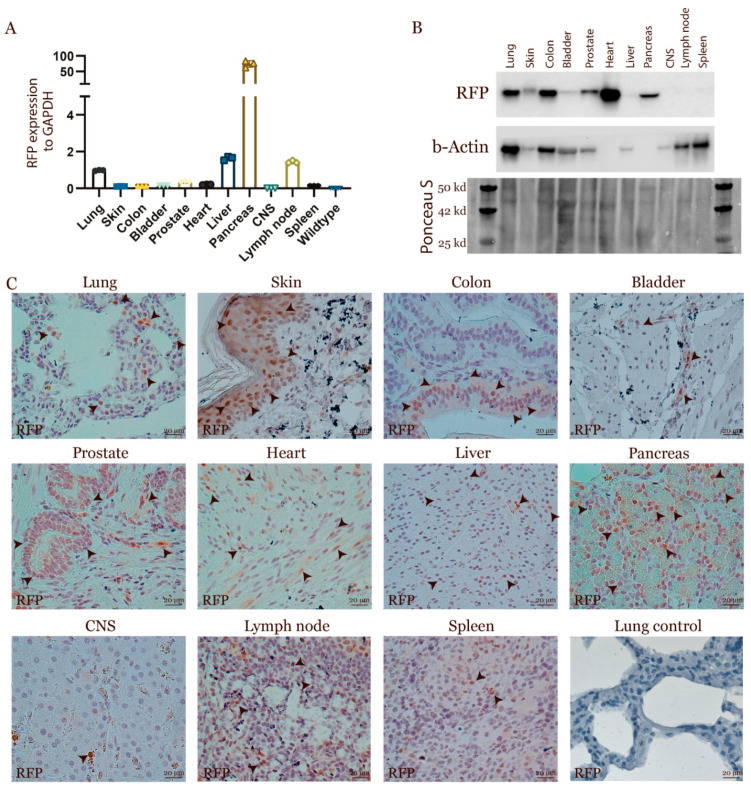
Expression profile of RFP in multiple organs. (**A**) Expression of RFP in multiple organs was assessed by qPCR. Expression was normalized to GAPDH, and the expression in lung tissues was set to 1 (*n* = 3). (**B**) Western blot RFP was performed on samples from multiple organs. Equal amount of proteins were loaded, and beta-actin and Ponceau S were used as loading control. (**C**) Tissue sections for different organs were stained with an antibody against RFP (brown stain). Arrowheads mark the positive cells. Haematoxylin staining was used to stain the nucleus of the cells (blue stain) (*n* > 3). Western Blot Images can be found in Appendix A.

**Figure 3 cancers-13-03024-f003:**
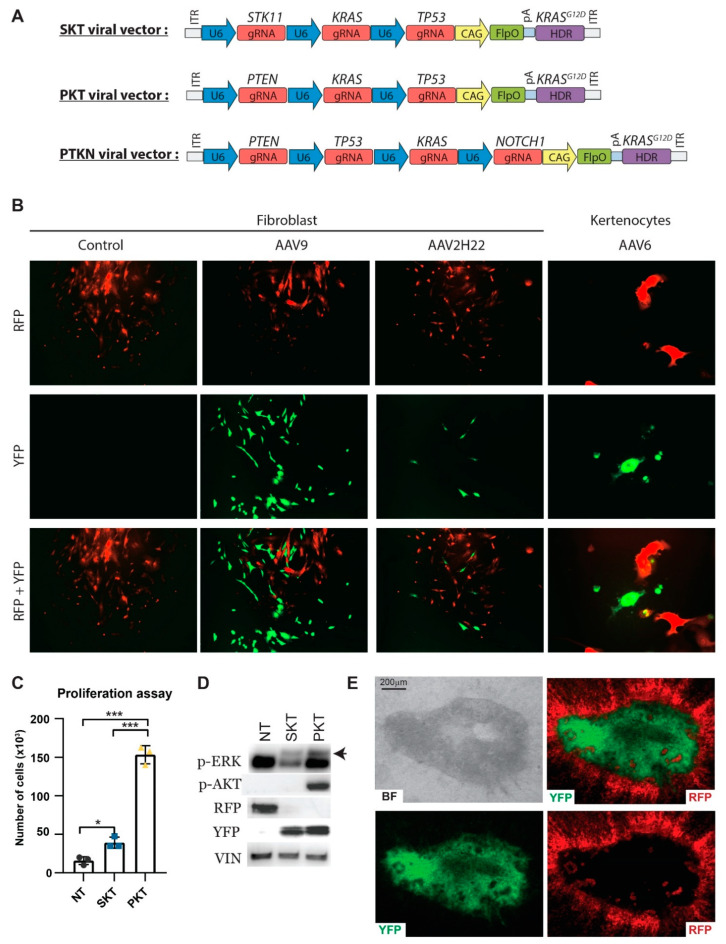
CRISPR-induced mutation in vitro. (**A**) Overview of the AAV viral vectors containing the FlpO gene used to activate the Cas9 expression, a KRASG12D repair template and sgRNAs against the following target genes: STK11, KRAS, TP53 (SKT). PTEN, TP53, KRAS (PKT). PTEN, TP53, KRAS, and NOTCH1 (PTKN). (**B**) Primary fibroblasts and keratinocytes were transduced with AAV particles, containing sgRNAs and FlpO expression. Transformed cells turn from red to yellow upon FlpO recombination. Images were taken after one passage of the cells. (**C**) Proliferation assay on non-treated, SKT_AAV2H22 or PKT_AAV9-treated Cas9 fibroblasts. Cells had undergone five passages before the proliferation was measured. * *p* < 0.05, *** *p* < 0.001. (**D**) Western blot on protein lysate from the proliferation assay. Black arrow marks p-ERK. Vinculin (VIN) was used as loading control. (**E**) Image 6 weeks after SKT_AAV2H22-treated air–liquid interface (ALI) Cas9 pig pseudostratified, ciliated, columnar tracheobronchial epithelial cells. Western Blot Images can be found in Appendix A.

**Figure 4 cancers-13-03024-f004:**
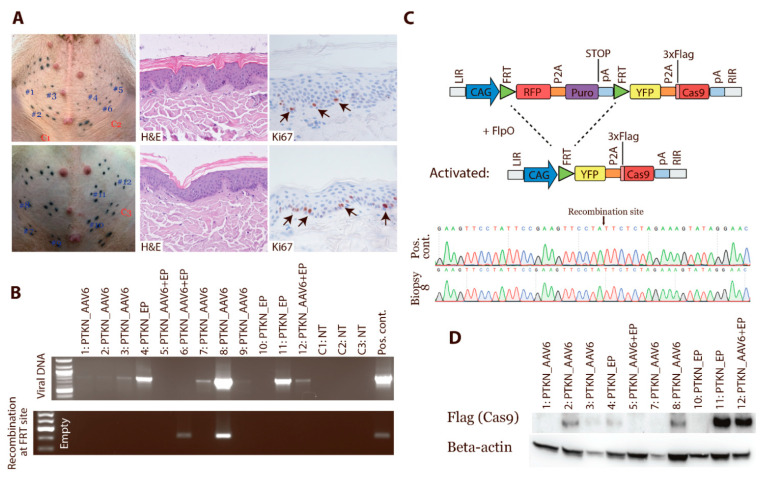
In vivo activation of Cas9 expression. Cas9 minipigs were transduced in multiple areas in the abdominal skin, and biopsies were taken after 1 month. (**A**) Tissue sections from transduced areas and control skin were stained for H&E and Ki67 (*n* > 3). (**B**) Biopsies were analysed for the presence of viral DNA and for recombination at the FRT sites and hereby activation of Cas9 expression. The PTKN construct was used as either plasmid or in AAV6 particles. Some areas underwent electroporation (EP) to enhance the delivery. NT: nontreated. (**C**) Two samples underwent Sanger sequencing to confirm recombination of the transposon at the FRT sites. (**D**) Proteins isolated from the biopsies were analysed by Western blotting for Cas9-Flag expression. Beta-actin was used as loading control. Western Blot Images can be found in Appendix A.

**Figure 5 cancers-13-03024-f005:**
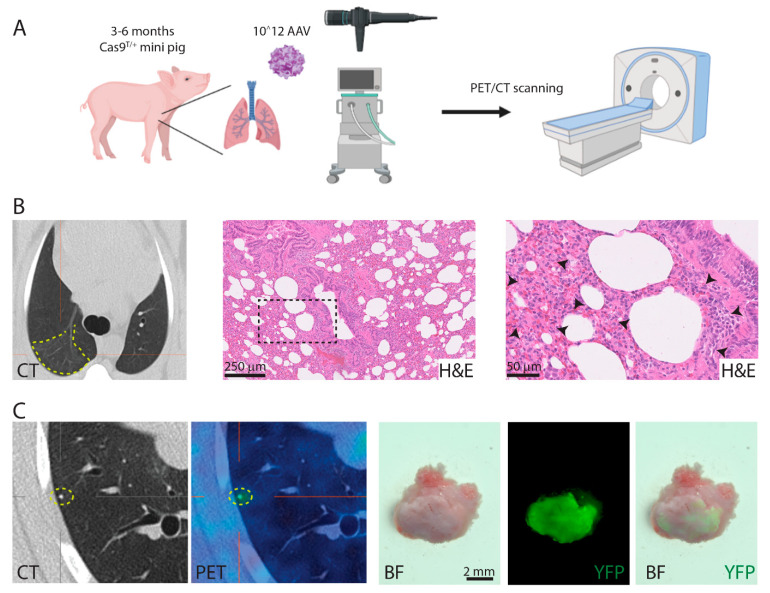
Delivery of AAV particles to the lungs. (**A**) Transgenic Cas9 pigs were transduced with 10^12^ AAV particles by aerosol or through a bronchoscope. The pigs were PET/CT scanned up to three times over 18 months after AAV transduction. (**B**) Inflammation in a lung lobe after AAV induction. Histology analysis revealed chronic inflammation by infiltrating lymphocytes (marked by black arrowheads). (**C**) PET/CT scanning shows increase FDG uptake in area of 5 mm^3^. Analysis of the spot revealed calcification with auto-fluorescent seen as YFP.

## Data Availability

Data is contained within the article or Appendix A.

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
