# Peer review of "The CRISPR/Cas9 Minipig—A Transgenic Minipig to Produce Specific Mutations in Designated Tissues"

_cancers, 2021, doi:10.3390/cancers13123024_

Round 1

Reviewer 1 Report

The authors report the generation and characterization of a Cas9 transgenic pig. To my knowledge this is the 3rd time such animals have been generated. The two previous publications, of which one also used minipigs, had placed the Cas9 gene via homologous recombination in the porcine ROSA26 locus. Cas9 expression was either ubiquitous or Cre inducible. Both showed Cas9 functionality in vitro and in vivo.

Novelty regarding the current manuscript:  transposon vectors were used for transgene integration, and Cas9 activation required Flp recombinase. Expression of the randomly integrated transgene could be confirmed via fluorescent marker genes and by mRNA, protein analysis and immunohistology. Cells isolated from the Cas9 transgenic pigs supported genome editing after addition of guide RNAs (gRNAs)

Similar to one of the previous publications gRNAs were designed to target tumor suppressor genes and to introduce an oncogenic mutation (G12D) into the porcine KRAS gene. Both publications aimed to induce lung tumorigenesis using either Lenti viral vectors (previous publication) or AAV (current publication) for delivery of the recombinase and the gRNAs. However, besides considerable efforts, the current publication failed to show in vivo genome editing in lung tissue. For skin they could prove Cas9 activation, but again no genome editing. Which might have been due to an immune reaction to cells expressing AAV or the Cas9 protein.

In summary: the publication shows a solid effort, the successful generation of Cas9 transgenic pigs and the functionality of a complex transgene in vitro. In vivo delivery methods still have to be improved to ensure functionality in vivo and to generate a porcine cancer model.

Prior to any publications the following points should be addressed:

  • Line 69 states “The cloning process is hampering generation of new transgene porcine models”. This should be rephrased, it might be inefficient, but it enables the generation of porcine models, as their own experiment shows.
  • Could the authors explain why they didn’t use cells from Göttingen minipigs for the nuclear transfer experiment, instead of backcrossing to Göttingen minipigs?
  • Line 239 Please explain “four ink stains”
  • Fig 1b Southern blot analysis: Please indicate the restriction sites used, in order to understand why multiple or single bands were detected. The figure legend mentions 5 copies for pig 41, the main text 6 copies. Please correct this.
  • 1c shows that not all cells had undergone Flp recombination, possibly only 50%, Why doesn’t the Western Blot (Fig.1d) reflect this? Cells used for the Western Blot are from a pig with how many copies?
  • 1c Why are the cells labeled as GFP and not YFP?
  • 2 a and b: Please explain the discrepancies between mRNA and protein expression in the various tissues (e.g. pancreas, heart).
  • 2 c RFP expression is not ubiquitious, is this correct? Did you detect any expression in lung tissue?
  • Line 368: What are non-activated fibroblasts? transfected with guides but not Flp?
  • 3 b and d. The fluorescent image shows presence of YFP and RFP positive cells after Flp recombination. The Western blot indicates 100% efficiency. please explain.
  • Line 442: if you do not see genome editing you can not conclude: “that targeting up to three genes simultaneously in the lung tissues of the pig is not 442 sufficient to induce cancer” see also line 506.
  • Line 460: should this read “applied the CRISPR/Cas system to the porcine cells”?
  • Line 471: the meaning of the following sentence is unclear: “Their data is resampling tumor formation in a Cas9 mouse model,..”
  • Line 502: ”we have developed a porcine utility model with global inducible Cas9 expression.” Please re-phrase utility model and global expression. The expression observed was very variable.
  • Language: some corrections are required e.g. porcine is an adjective and should not be used when you mean pig. “Cas9 transgene pigs” should read transgenic pigs.

Author Response

We thank the reviewer for thorough and constructive comments on this manuscript. We have addressed the questions and comments and we hope the reviewer finds our answers appropriate. We are happy to elaborate further if required. 

  • Line 69 states “The cloning process is hampering generation of new transgene porcine models”. This should be rephrased, it might be inefficient, but it enables the generation of porcine models, as their own experiment shows.

AW: We agree with the reviewer and have changed the sentence. Line 69: “Notably, cloning is inefficient and only a few percent of the implanted blastocysts/morulae will result in viable piglets.”

  • Could the authors explain why they didn’t use cells from Göttingen minipigs for the nuclear transfer experiment, instead of backcrossing to Göttingen minipigs?

AW: At the beginning of the project, a license to use cells from Göttingen minipigs could not be obtained. Therefore, we used a mix race of minipigs for the cloning.

  • Line 239 Please explain “four ink stains”

AW: We used tattoo ink to mark the area of injections. It can be seen on figure 4a. We have added the word “dots by tattoo ink” to line 241: “Each treated area was marked with four dots by tattoo ink stains, to locate the treated areas at the end of the study.”

  • Fig 1b Southern blot analysis: Please indicate the restriction sites used, in order to understand why multiple or single bands were detected. The figure legend mentions 5 copies for pig 41, the main text 6 copies. Please correct this.

AW: We used the restriction enzyme XmaJI for digestion of the DNA (mention at line 205). As we have used transposon for random integration to the genome, we were not able to predict the band sites on the SB. This gives problems, as two bands with similar site appear as one band. For pig 40 and 44, we see two bands (two copies of the transposon) even though they have nearly similar size.

For pig 41, 5 bands are present but the band at 6 kb is more intense and we predict that we have a double band even though we did not have perfect separation. We therefore stated, that clone 41 has 6 copies.    

  • 1c shows that not all cells had undergone Flp recombination, possibly only 50%, Why doesn’t the Western Blot (Fig.1d) reflect this? Cells used for the Western Blot are from a pig with how many copies?

AW: We took the pictures of the cell at passage two after transfection. Here we still see both transformed cells and non-transformed. The WB is made on cell lysate from passage 7, here the transformed cells had overgrown the non-transformed cells. The transformed cells had a higher proliferation index, as seen in figure 3c. We have added the cell passage number to the figure legend to clarify this point.

All the in vitro analysis has been performed on cells obtained from pigs with one copy. However, we have also used pigs with two copies for in vivo experiments, to address if we will obtain different results.  

  • 1c Why are the cells labeled as GFP and not YFP?

AW: This is a mistake. We see the YFP under the GFP filter, but it is YFP that is detected.

  • 2 a and b: Please explain the discrepancies between mRNA and protein expression in the various tissues (e.g. pancreas, heart).

AW: We do see discrepancies between mRNA, WB and IHC for RFP expression. For RNA and WB we see different expression in the house-keeping gene, which influences the normalization of RFP expression. We have mentioned this in the discussion on line 477 to 483.

  • 2 c RFP expression is not ubiquitious, is this correct? Did you detect any expression in lung tissue?

AW: It is correct that the expression is not ubiquitous and that different levels can be detected in the cells. This could suggest that expression is variable to some extent. We have added arrowheads to figure 2c, to mark positive cells. We did see expression in the lung tissues, which we have estimated to 25%. We have mentioned that on line 481 to 487.

  • Line 368: What are non-activated fibroblasts? transfected with guides but not Flp?

AW: These cells have been transfected with an empty plasmid. We have now added that to the text on line 329: “The genetic design was validated by transfecting transgenic Cas9 fibroblasts with a FlpO and sgRNA expressing plasmid or an empty plasmid. Cells transfected with an empty plasmid appeared red….”

  • 3 b and d. The fluorescent image shows presence of YFP and RFP positive cells after Flp recombination. The Western blot indicates 100% efficiency. please explain.

AW: Here we have performed the experiment as in figure 1. Pictures were taken at passage one and the WB were performed on cell lysate from passage 5. We have added this to the figure legend.

  • Line 442: if you do not see genome editing you can not conclude: “that targeting up to three genes simultaneously in the lung tissues of the pig is not 442 sufficient to induce cancer” see also line 506.

AW: We agree with the reviewer. As we have not confirmed CRISPR induces mutation to the lung, we cannot conclude that those three genes were mutated. We have rephrased the sentence on line 442 and removed the statement from the conclusion at line 506. Line 444: “In summary, these analysis revealed that targeting up to three genes simultaneously by AAV delivery of sgRNA to the lung tissue, was not sufficient to induce cancer over a period of 18 months.”

  • Line 460: should this read “applied the CRISPR/Cas system to the porcine cells”?

AW: Yes. We thank the reviewer the correction.

  • Line 471: the meaning of the following sentence is unclear: “Their data is resampling tumor formation in a Cas9 mouse model,..”

AW: We have corrected the sentence and added this: “Cancer induction in their pig model resembles tumor formation in a similar Cas9 mouse model…”

  • Line 502: ”we have developed a porcine utility model with global inducible Cas9 expression.” Please re-phrase utility model and global expression. The expression observed was very variable.

AW: We agree with the reviewer, that this statement can be misunderstood. We have now rephrased: In conclusion, we have developed a porcine model with inducible Cas9 expression in the majority of organs.

  • Language: some corrections are required e.g. porcine is an adjective and should not be used when you mean pig. “Cas9 transgene pigs” should read transgenic pigs.

AW: After suggestion from both reviewers, the manuscript has undergone language correction by a native speaking scientist.  

Reviewer 2 Report

This manuscript describes the generation of Cas9 expressing minipigs. The authors use a clever strategy of transfection of Sleeping Beauty transposons containing Cas9 into minipig fibroblasts, followed by HMC nuclear transfer to generate transgenic embryos. These experiments are nicely detailed, as is the characterization of the resulting transgenic mini pigs. This includes a detailed description of the extent and level of Cas9 expression in distinct tissue types. These mini pigs represent a highly useful reagent for the broader cancer community.

The authors then use AAV-mediated delivery of gRNAs to disrupt cancer genes and demonstrate that Cas9 expressing fibroblasts that are targeted with gRNAs that induce KRASG12D, as well as tumor suppressor inactivation, can enhance cell proliferation.

The generation of Cas9 expressing minipigs is not completely novel, but this specific approach is unique and considerably adds to the cancer models available in this important model organism. Thus, I support publication of this work, but have one significant concern. The authors state that "In summary, these analyses revealed that targeting up to three genes simultaneously in the lung tissues of the pig is not sufficient to induce cancer over a period of 18 months." They then attempt to reconcile their findings with another published work showing cancer induction in Cas9 expressing mini pigs bearing gRNAs targeting tumor suppressors. However, the authors fails to show that they can effectively induce changes in all three target genes. The cell culture proliferation experiment shows that one or more of these cancer genes is modified and the supplementary data shows that all of the target genes are modified to some extent in a subset of cells. However, we do not know whether there are any cells in vivo that have successful editing of both alleles of each tumor suppressor (in addition to a single allele of Ras). Thus, it is difficult to conclude that editing of three genes is insufficient to promote cancer (which is a very important point!).

Either the authors need to demonstrate that all of the target genes are edited in a subset of cells or that need to alter their conclusions to introduce the possibility that incomplete editing may underlie the lack of cancer phenotypes.

Author Response

We thank the reviewer for comments on this manuscript. We have addressed the question and hope the reviewer find our answer appropriate. We are happy to elaborate further if required. 

  • Either the authors need to demonstrate that all of the target genes are edited in a subset of cells or that need to alter their conclusions to introduce the possibility that incomplete editing may underlie the lack of cancer phenotypes.

AW: We agree with the reviewer, as we have not confirmed CRISPR induces mutation in the lungs, we cannot conclude that those three genes were mutated. We have rephrased the conclusion:

“In conclusion, we have developed a porcine model with inducible Cas9 expression in the majority of organs. We validated the model in vitro in fibroblasts, keratinocytes, and pseudo-stratified lung epithelial cultures and in vivo in the skin of the minipigs by Cas9 construct activation through AAV transduction. Thus, an in vitro and, potentially, an in vivo model of human cancer has been generated in a minipig for pre-clinical studies.”